# Measurements and modeling of water levels, currents, density and wave climate on a semi-enclosed tidal bay: Cádiz (SW Spain)

Carmen Zarzuelo[1], Alejandro López-Ruiz[1], María Bermúdez[2], and Miguel Ortega-Sánchez[2]

[1]Departamento de Ingeniería Aeroespacial y Mecánica de Fluidos, Universidad de Sevilla, Camino de los Descubrimientos s/n, 41092, Seville, Spain

[2]Andalusian Institute for Earth System Research, University of Granada, Avda. del Mediterráneo, s/n, 18006 Granada, Spain

**Correspondence:** Carmen Zarzuelo (czarzuelo@us.es)

**Abstract.** Estuarine dynamics are highly complex as a result of the temperature and salinity gradients, as well as the multiple interactions between atmospheric, maritime and hydrological forcing agents. Given the environmental and socioeconomic importance of estuaries and their current and future threats due to human interventions and climate change, it is of vital importance to characterize these dynamics, monitor their evolution and quantify the expected impacts derived from climate change. This paper presents a hybrid database combining data obtained in six field surveys (in 2012, 2013 and 2015) and results from a physically-based 3D numerical model for the Bay of Cádiz (southern Spain), a highly anthropized mesotidal estuary. The 3D dataset includes water levels, currents, density and wave climate, allowing for an analysis of bay dynamics at different time scales ranging from intratidal processes to seasonal variabilities. The results offer an example of the potential uses of the dataset and include (1) an assessment of the spatial and seasonal variability of the estuarine dynamics and (2) an analysis of the effects of severe weather events. These examples provide convincing evidence regarding how the dataset can be employed in multiple research fields and applications, including ocean-bay interactions, water exchange between basins, long- and short-wave propagation along creek systems and energy extraction of tidal waves. Therefore, this hybrid dataset may be of significant interest for stakeholders and scientists from different sectors (water engineering, ecology, urban development, energy, etc.) working on the environmental management of the Gulf of Cádiz and other tidally-dominated shallow bays. It can also serve as a benchmark test for numerical hydrodynamic models, infrastructure intervention assessments (e.g., dikes or breakwaters) or renewable energy conversion system models.

## 1   Introduction

An estuary is a partially enclosed coastal water body where freshwater from rivers and streams mixes with saltwater from the ocean (Alahmed et al., 2022). Although influenced by tides, estuaries are frequently protected from the full force of ocean waves, winds and storms (Eryani and Nurhamidah, 2020). The sheltered waters of estuaries also support freshwater and saltwater marshes, swamps, sandy beaches, mud and sand flats, rocky shores, oyster reefs, mangrove forests, river deltas, tidal pools and seagrass beds, which increase the diversity of the environment (Hopkinson et al., 2019; Hobohm et al., 2021). Moreover, wetland plants and soils also act as natural buffers between the land and ocean, absorbing flooded waters and dissipating storm surges (Asari et al., 2021). The protected coastal waters of estuaries also support important public infrastructure, serving

as harbors and ports vital for shipping and transportation, which must be carefully managed to ensure the sustainability and protection of this natural resource (Allison et al., 2020).

The complexity and the environmental and socioeconomic importance of the estuaries described above highlight the need for a detailed characterization of the dynamics of these coastal systems. This information will be even more important in upcoming years and decades, considering the expected impacts of sea level rise due to global warming and climate change in these areas (Haasnoot et al., 2019). Currently, there are three main approaches for characterizing estuaries: (1) physical models, (2) field data, and (3) physically-based numerical modeling. Physical models replicate coastal systems at a reduced scale so that the major dominant forces acting are reproduced (Reeve et al., 2018). Although these models are extensively applied in the design of major hydraulic engineering works and greatly enhance our understanding of fluvial, estuarine and coastal processes (Weisscher et al., 2020), the characteristics of estuarine dynamics complicate their use; complex geometries, a significant number of different forcings acting simultaneously, and water density gradients have to be considered. Furthermore, the conditions required to appropriately model the substrate, sediment and biota are difficult to reproduce in the laboratory.

Field data are obtained through multiparameter and multiscale environmental monitoring at a limited number of discrete locations where appropriate scientific instrumentation can be deployed (Garel and Ferreira, 2015). With this approach, hydrodynamic variables and water properties such as water levels, currents, salinity and/or water density can be measured. The usefulness of these data in characterizing estuarine dynamics relies primarily on the design of the deployment (type, number, location and setup of the equipment), the length of the survey, and the representativeness of the meteorological, atmospheric, maritime and hydrological conditions during the selected period of time. Despite providing the most realistic characterization of the estuarine dynamics currently available, field data have severe limitations due to the isolated locations of the environmental monitoring devices and the limited variables and conditions that can be measured.

Physically-based numerical models overcome these limitations by resolving the conservation equations that are approximations of laws for water flow and/or sediment transport (e.g., Meyer-Peter and Müller (1948); Van Rijn (2007); Baar et al. (2019)). These equations are solved in a spatial domain that can cover complete estuarine areas with spatial resolutions up to tens of meters. The interest in such models lies not only in the transition from isolated to spatially distributed data but also in the analysis of conditions that did not occur during the observation period, including prospective conditions resulting from a changing climate (e.g., Yang et al. (2015); Del-Rosal-Salido et al. (2021)), as these models enable full control of the initial and boundary conditions (Weisscher et al., 2020) in contrast to field data. The main limitation of numerical modeling comes from the significant number of model parameters (e.g., bed friction, viscosity, diffusivity) that have to be adjusted to adequately reproduce the hydrodynamic and transport processes. This problem is solved by calibrating and testing these parameters using field data and exploring the sensitivity to changes in their values (van Maren and Cronin, 2016). Hence, field data and numerical modeling are complementary approaches that can be used simultaneously to analyze estuarine dynamics, to adequately reproduce the processes observed in nature, and in turn to allow prognostic and extreme event analysis.

This paper presents a hybrid dataset developed to characterize the hydrodynamics, salinity and temperature distributions in the mesotidal estuary of the Bay of Cádiz (southern Spain). This bay is a paradigmatic example of a complex geometry, tidally-dominated bay where different basins are connected through narrow channels and waves and river discharges have limited

influence. Similar estuaries can be found worldwide, such as San Francisco Bay (Gartner and Walters, 1986), Jiaozhou Bay (Li et al., 2014), the St Lucia estuarine lake (Schoen et al., 2014), and the Sylt-Romo Bight (Purkiani et al., 2016). The dataset includes both field data and numerical modeling results that were obtained for the study area during the last decade. In addition to analyzing the general hydrodynamics and transport processes in the bay (Zarzuelo et al., 2021), the dataset was used to study the impact of human interventions on estuarine dynamics (Zarzuelo et al., 2015) and the tidal energy potential in the area (Zarzuelo et al., 2018). With the description presented here and its open-source publication, the dataset aims to provide treated and formatted hydrodynamic data for engineers and researchers working on the environmental management of the Gulf and Bay of Cádiz, including aspects such as water quality (Jiménez-Arias et al., 2020; Besada et al., 2022), ecosystem functioning (Miró et al., 2020; Haro et al., 2022) or energy assessments (Zarzuelo et al., 2018; Legaz et al., 2020). These data may also be of interest to modelers; in particular, they can be used as a benchmark test case for numerical hydrodynamic models, as a reference for human intervention alternatives (e.g., piles, dikes or breakwaters), or as a basis for modeling renewable energy conversion systems. In addition, the dataset may be of interest for scientists aiming to compare the behavior of different estuarine areas.

## 2    Area description

The Bay of Cádiz is a mesotidal, mixed energy and well-stratified estuary (Zarzuelo et al., 2015, 2017, 2020) in southern Spain, characterized by a temperate climate. The bay consists of three areas (A, B and C in Fig.1) and a zone of tidal flats and marshes, with a total extension of 140 km2. The outer basin (A), with an area of 70 km$^2$, is characterized by mild slopes and depths ranging between 5-10 m. This area is the most exposed to wave action due to its opening to the Atlantic Ocean and is also connected with the estuaries of the San Pedro and Guadalete rivers. The inner basin (C) has an extension of 50 km2, with intertidal areas that give rise to an area of gentle slopes, which is reconnected to the Atlantic Ocean through the 17 km-long tidal creeks of Carracas and Sancti-Petri (Zarzuelo et al., 2018, 2020). Finally, the Puntales channel (B) connects the outer and inner basins with dimensions of 3.1 km and 1.7 km in length and width, respectively. This latter area exhibits the greatest depths (up to 18 m) and corresponds to the zone where most recent human interventions are carried out. The bay has tidal ranges of approximately 2-4 m (neap and spring tides, respectively) and a moderate wave climate that propagates mainly from the Atlantic Ocean with a limited incursion inside the bay. The influence of freshwater discharges is weak since the mean discharge rates are approximately 20 m$^3$/s compared to a tidal prism of $10^5$ m$^3$/s (Zarzuelo et al., 2015, 2017). With these characteristics, the Bay of Cádiz is considered an example of tidally-dominated, shallow, semi-enclosed bays (Kitheka, 1997; LEE et al., 1997; Newton et al., 2014; Shang et al., 2019).

The estuary contains extensive areas of intertidal sand and mudflats that support a variety of characteristic benthic fauna depending on the nature of the substrate. In its inner basin and around the creeks of Carracas and Sancti-Petri, there are also extensive areas of mudflats. This space, dominated by swamps, marshes, and salt flats, is characterized by an astounding wealth of biodiversity, with a long historical tradition of salt mining. This area has undergone a major transformation due to progressive sedimentary infill conditioned by the important biosedimentary role of vegetation, which often controls the morphological evolution of these tidal environments (D'Alpaos, 2011). At the same time, a significant part of the emerged

marshes within the Bay of Cádiz has been almost completely anthropized over the past 60 years (Gracia et al., 2017). In recent years, many of the structures of old salt marshes have disappeared as a result of the intense transformation that is being carried out in the area.

Since the twentieth century, human interventions such as ports, bridges or salt marshes have been developed to improve the economic and social activities within the bay and its surroundings. The vast majority of these interventions are located in the Puntales channel (Zarzuelo et al., 2015), interfering with the exchange of water, heat and sediment between the outer and inner basins (Zarzuelo et al., 2021).

## 3 Methods

To analyze the dynamics of the Bay of Cádiz, two methods were used: (1) a combination of 3 bottom-mounted and 3 vessel-towed field surveys that provided real data with high temporal resolution and (2) a 13-month 3D numerical simulation that was calibrated and tested with the field data. This latter simulation provides high spatial resolution data for a longer time period for which variabilities down to seasonal scales can be analyzed. The combination of both methods allows us to analyze the hydrodynamic and morphodynamic behavior of the Bay of Cádiz on a global scale, which was the main milestone of the Project P09-TEP-4630 (funded by the Andalusian Regional Government) that financially supported the study to obtain this hybrid database.

### 3.1 Design of the field surveys

More than 20 instruments during 6 different surveys were used to measure water levels, currents, water density and conductivity, and wave conditions. The total length of the records considering all the surveys exceeds 8 months. The two types of field surveys designed, bottom-mounted and vessel-towed, are described within the next sections.

#### 3.1.1 Bottom-mounted observations

As shown in Table 1, three different types of instruments were used in these surveys: ADCPs, CTs and tide gauges. The ADCPs are Nortek Aquadopp Profiler AST, which are acoustic Doppler current profilers with the added feature of acoustic surface tracking (AST). It emits sound pulses in three directions near the vertical, and receives the echoes from suspended particles drifting with the current. The compass is calibrated directly on the ship using a track crane to avoid damaging the surrounding metal. It is then placed in the water and the divers check that the "x" direction coincides with the East. The velocities obtained from different particles are then averaged within subdivisions of the water columns (cells). The technical specifications of the Nortek ADCP are: acoustic frequency 0.6MHz (1 device), 1 MHz (2 devices) and 2 MHz (3 devices), velocity range $\pm$ 10 m/s, accuracy (water velocity) 1% of measured value, $\pm$ 0.5 cm/s and maximum range 40 m, 25 m and 10 m, respectively. The ADCPs measured surface pressure, bottom temperature and velocities along the entire water column. From these measurements, the energy spectrum was assessed to determine the wave height, peak period and wave direction. All acoustic signals were emitted from three beams inclined at 25° from the vertical and equally spaced at 120°. The ADCPs

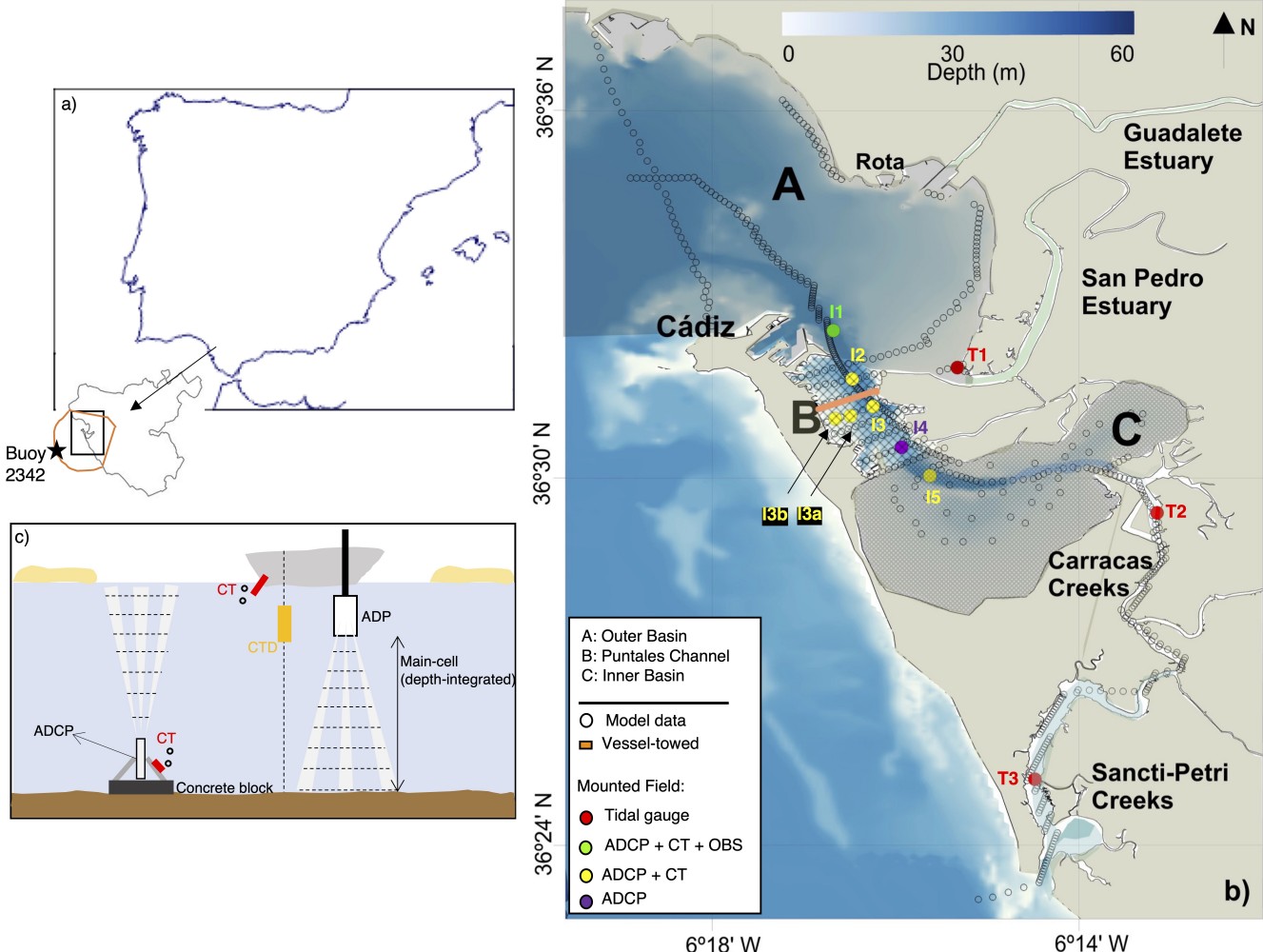

**Figure 1.** Location (a-b) and description (c) of the monitoring system. The Bay of Cádiz (a) is located in the southern Iberian Peninsula. The model domain is indicated by the orange line (a). The locations (b) of the model data are shown as empty circles. The locations of the observed data stations are indicated by coloured circles (colour legend corresponds to the instrumentation used in each location): Field survey2012 (I1-I5; T1-T3), Field survey2013 (I3; I3a; I3b) and Field survey 2015 (I3; I5) (see Table 1 for more details on sampling periods). Station (c) is equipped with a multiparametric probe (CT and CTD) and an acoustic Doppler current profiler (ADP). Salinity, temperature, wave and wind data were obtained from buoy 2342 (Puertos del Estado, Spanish Ministry of Public Works). Average discharges of the Guadalete and San Pedro rivers and daily averaged data of solar radiation, air temperature and humidity, and cloudiness were provided by the Andalusian Regional Government. Bathymetric data were provided by the Instituto Hidrográfico de la Marina (Spanish Ministry of Defense) and the Bay of Cádiz Port Authority.

included a high-resolution pressure sensor, as well as compass/tilt and temperature sensors. The distance from the base of the
125 concrete block (tripod) to the top of the ADCPs was 0.3 m (0.8 m). The three current velocity components (east, north, and

vertical) were measured in cells of varying thickness (hereafter referred to as multicell data) along the water column (Table 1). In addition, the depth-integrated velocities were measured in one cell (main cell data, hereafter) the vertical extent of which was automatically adjusted near the surface based on pressure records. Both the main and multicell measurements started at a blanking distance above the instrument and thus at different distances above the bottom (see Table 1). The number of cells

was always sufficient to cover the entire water column above the instrument. In addition, this distance varied over time due to changes in bed elevation, especially those caused by the burial and tilting of the mooring structure (Lobo et al., 2004; Morales et al., 2006; Garel and Ferreira, 2015).

The CTs are SBE 37-SMP pumped MicroCAT, with a high-accuracy conductivity and temperature (pressure optional) recorder with Serial (RS-232) interface. Technical specifications are: power supply 7.8 Amp-hour, resolution 0.00001 S/m

(conductivity) and 0.0001 °C (temperature), accuracy (conductivity) $\pm$ 0.0003 S/m (0.003 mS/cm), accuracy (temperature) $\pm$ 0.01 °C (35 °C to 45 °C), $\pm$0.5 cm/s and maximum range 7 S/m (conductivity) and 45 °C (temperature). The conductivity-temperature (CTs) sensors measured bottom temperature and conductivity (see Table 1), which are converted to salinity values. They recorded data every 15 minutes to match the ADCP data. The OBS-3+ is a submersible turbidity probe with side-facing optics. It uses OBS technology to measure suspended solids and turbidity. The technical specifications of the OBS-3+ are:

optical power 2000 $\mu$W, infrared wavelength 850nm $\pm$ 5nm , accuracy (turbidity) 2% of reading or 0.5 NTU, accuracy (concentration) 2% (4%) of reading or 1 mg/l (10 mg/l) for mud (sand) and maximum range 10000 mg/l (mud) and 100000mg/l (sand). The OBS measured turbidity at the bottom; it was connected to an ADCP and measured with the same frequency, burst interval and period, sharing its configuration with the ADCP as they were nested. Finally, the tide gauge model TD304 from SAIV A/S is a high precision instrument for recording/monitoring tides or water levels (tide gauge). The technical specifica-

tions are: response time 0.1 s, resolution 0.0001 dbar (m), accuracy (water level) $\pm$ 0.01% FS and maximum range 1000 m. The tide gauges only measure pressure, with 10 sec bursts every 15 min (1 Hz). For more details on the configurations of each field survey, see Table 1.

Three field surveys using bottom-mounted instruments were carried out. These surveys were designed to have short- to mid-term time periods with real data that could (1) characterize the dynamics of the bay and (2) calibrate the numerical model

considering a wide range of different tidal, wave, wind and weather conditions. The three deployments were as follows:

– Mounted *Field2012* (22-12-2011 to 22-05-2012): Twelve instruments were anchored in 8 positions (I1-I5; T1-T3) distributed along the bay and inside the creeks (see Table 1 and Fig.1b). The instruments included 5 ADCPs, 1 OBS, 3 CTs and three tide gauges. Concrete blocks were used to secure the instruments to the seabed and avoid displacements. This initial survey was designed to characterize the dynamics of the three main areas of the bay (A, B and C in Fig.1b) and

the tidal creeks. The data were also used to assess the balance and water exchange between the basins and the creeks, and to characterize the effects of extreme wind and wave events, which are common during the winter season, on the dynamics of the bay.

– Mounted *Field2013* (25-05-2013 to 28-08-2013):In this second survey, 6 instruments (3 ADCPs and 3 CTs) were moored in 3 locations distributed along a cross-section of the Puntales channel to characterize the water exchange between the

outer basin and the channel itself (see Table 1 and zone B in Fig.1b). In this case, metal tripods were used to secure the instruments on the seabed. After Field2012, it was observed that the main exchange section was the connection between the outer basin and the Puntales channel. Therefore, this survey was designed with 3 locations evenly distributed along a section just at the entrance of the channel.

– Mounted *Field2015* (16-09-2015-23-09-2015): Finally, since the results provided by the numerical model suggested that the bay behaves as an inverse estuary, 4 instruments (2 ADCPs and 2 CTs) were anchored with tripods for stability in 2 locations to observe heat exchange differences and characterize the water exchange between the Puntales channel and the inner basin (Table 1 and Fig.1b).

### 3.1.2 Vessel-towed observations

As a result of the first bottom-mounted survey (Field2012), important exchange flows were identified between the Inner and Outer basins through the Puntales channel. To analyze these flows in detail, three vessel-towed continuous surveys were conducted using two different devices (ADP and CTD).

The ADP used for the vessel-towed field surveys is a Sontek YSI 1 MHz acoustic Doppler profiler with a maximum profiling range of 35 m. The velocity range is $\pm$ 10 m/s, the accuracy $\pm$ 1% of the measured velocity, $\pm$ 0.5 cm/s and a resolution of 0.1 cm/s. Underway velocity profiles were recorded with a boat-mounted ADP in 50 cm bins with 120 pings per ensemble during a complete tidal cycle. Navigation was performed with a global positioning system (GPS). In 2013, a total of 23 neap tides transects and 24 spring tides transects were sampled, whereas in 2015 a total of 25 neap tide transects were sampled. The transect data were organized in vertical and horizontal equal grids across the channel.

The CTDs used are the SBE 19plus V2 SeaCAT, which measures conductivity, temperature, and pressure at 4 scans/sec (4 Hz) and provides high accuracy and resolution. Technical specifications are: resolution 0.00005 S/m (conductivity), 0.0001 °C (temperature) and 0.002% of full scale range (pressure), accuracy (conductivity) $\pm$ 0.0005 S/m, accuracy (temperature) $\pm$ 0.005 °C, accuracy $\pm$ 0.1% of full scale range (pressure) and maximum range 9 S/m (conductivity), 35 °C (temperature) and 350 m (pressure).

The three vessel-towed continuous surveys were conducted in 2013 (2) and 2015 (1) during a tidal cycle at a cross-section of this channel:

– 2013: Two 13 h long acoustic Doppler profiler (ADP) surveys were performed during neap (7 July, Neap2013) and spring (22 August, Spring2013) tides across the Puntales channel (orange line-Fig.1b) to capture the seasonal variability and the water exchange between the outer basin and the channel. In addition, a conductivity-temperature-depth instrument (CTD SeaBird SBE19+, 4 samples at 1 Hz), plus a Wetlabs fluorometer and OBS turbidometer, mounted on a towed Guildline MiniBat undulator, enabled rapid sampling at five extra locations across the transect. Towing at approximately 5 knots provided vertical profiles roughly spaced 200 m every 2.2 hours. Samples of water were also taken to calibrate the sediment suspension in every CTD profile.

 One 13 h long survey was performed during neap (22 September, Neap2015) tides across the same transect (orange line-Fig.1b) defined in 2013 to capture possible changes produced by the new bridge (seaward of orange line-Fig.1b) built in the Puntales channel between 2007 and 2015 and to assess the temporal variations between 2013 and 2015.

### 3.1.3 Post-processing data

Once the data had been exported from the instruments, post-processing was used to remove errors. First, the data was checked for battery life. Then the compass ($<0.1°$) was corrected by comparison with the heading, pitch and roll ($<1°$), which cause deviations in the directions of the variables. The instrumentation was also checked for any rotation that could distort the information. Finally, a comparison was made between the pressure data and the water column in order to remove the cells that remained above the free surface (or to eliminate the influence of atmospheric pressure).

Assessing the quality of Acoustic Doppler measurements is essential to ensure reliable interpretation of velocity results. Errors caused by low signal strength, Doppler noise, signal aliasing and other effects such as side-lobe interference can introduce inaccuracies into the data. Such errors can make the data inaccurate, leading to increased data variance, mean bias and altered energy spectrum results. All instruments were calibrated prior to configuration (beam calibration, compass calibration, pressure offset) according to the manufacturer's recommendations. Processing steps included removal of poor quality data points using the manufacturer's Ocean Contour software. Data were corrected for magnetic declination, low correlation, side lobe interference, minimum amplitude and maximum amplitude spike, removing a total of 12% of the data, leaving 88% of the validated ADCP data. Data points flagged as poor quality were then replaced with NaNs in MATLAB ($\approx 20\%$).

The pressure signals were then corrected for variations in the atmospheric pressure. After sieving the data, the pressure was converted to free surface elevation following Nielsen (1989) The spectra and the directional spectrum were calculated to obtain the significant wave height, peak period and wave direction (O'brien, 1993) In this procedure, the nonlinear wave portion is low-pass filtered from the original bottom pressure data prior to estimating the surface wave values. The filter cutoff is determined for each wave burst after fitting a Pearson spectrum.

Temperature and salinity CTD data were post-processed using standard Seabird software and MATLAB routines. At this stage, peaks were removed, 1 dbar averages were calculated, and the downcast profiles of temperature and salinity were corrected using regression analysis. Once the salinity is known, the density is calculated. Suspended sediment concentration (SSC) was derived from OBS measurements (e.g., Downing (2006)) via calibration with suspended matter concentrations gravimetrically determined from water samples. The OBS calibration was accomplished by cross-correlation between the instrument readings and the SSCs obtained from water samples. Calibration was carried out at the laboratory of the University of Granada. Sediment was collected from the seabed and mixed in a mixing tank. The sediment concentration was slowly increased to obtain the relationship between voltage and sediment concentration.

The vessel-towed ADP data were processed to correct for heading errors and then averaged over of 1 min and 1 m depth intervals to provide higher spatial resolution. To mitigate the effects of side-lobe interference, the lower 10% of the velocity profiles of the signatures were removed. The depth at each location was estimated from the signature data. As the instruments were surface mounted, the location of the seabed was approximated to the nearest cell location by locating the prominent rise

in the return signal strength. The validity of the results was also assessed by the quality of the return signals. The consistency of the return signals within the sample volume over the sample period is measured by the percentage correlation. The amplitude is a measure of the return signal strength, while the Signal-to-Noise Ratio (SNR) indicates the strength of the return signals compared to the instrument noise (Doppler Noise). As a quality control measure, unusable vector data are identified in this study as having a correlation of <70% or an SNR of <10dB. For the signature data, a correlation threshold of <50% and an amplitude threshold of <30 counts was applied to identify invalid data. The ADP time series were processed for outliers, and small gaps were filled by cubic spline interpolation. Data sampled more frequently than once per hour were averaged to 1-h intervals. Currents were rotated into east–west and north–south basin components. Axis directions were determined by examining the principal axes of spatial variability and the orientation of the local bathymetry.

## 3.2 Numerical modelling

The 3D numerical modeling was performed with DELFT3D, a model developed for the coupled simulation of long and short wave dynamics, sediment transport and heat fluxes. Its subgrid approach for high-resolution bathymetric representation on structured grids (Lesser et al., 2004) makes it suitable for the simulation of hydrodynamics in complex estuaries such as the Bay of Cádiz. The model, fed with tidal harmonics, wind forcing, wave climate and heat fluxes, provides water levels, currents, density and wave climate with high spatial (both horizontal and vertical) and temporal resolution.

### 3.2.1 Model description

Long waves in Delft3D are modeled by the Flow module. This module solves the Navier-Stokes equations for an incompressible fluid under the shallow water and Boussinesq assumptions in two (depth-averaged) or in 3D using a finite difference scheme on staggered, curvilinear and well-structured grids to simulate the flow resulting from tidal and meteorological forcing. In 3D simulations, the vertical grid is defined using the $\sigma$ coordinate method. The continuity equation takes into account the kinematic boundary conditions at the surface and bottom:

$$\frac{\partial \eta}{\partial t} + \frac{\partial (hu)}{\partial x} + \frac{\partial (hv)}{\partial y} + \frac{\partial w}{\partial z} = hQ \tag{1}$$

where $\eta$ is the water level, $t$ is time, $h = z_b + \eta$ is the water depth with $z_b$ the bed level (positive below the mean sea level), $u$, $v$ and $w$ are the velocity components along the $x$, $y$ and $z$ coordinates, respectively, and $Q$ is represents the contributions per unit area of the sinks and sources of water mass. The momentum equations in $x$ and $y$ are:

$$\frac{\partial u}{\partial t} + u\frac{\partial u}{\partial x} + v\frac{\partial u}{\partial y} + \frac{w}{h}\frac{\partial u}{\partial z} - fv = -\frac{1}{\rho_0}P_x + F_x + \frac{1}{h^2}\frac{\partial}{\partial z}\left(\nu_v\frac{\partial u}{\partial z}\right) + M_x \tag{2}$$

$$\frac{\partial v}{\partial t} + u\frac{\partial v}{\partial x} + v\frac{\partial v}{\partial y} + \frac{w}{h}\frac{\partial v}{\partial z} + fu = -\frac{1}{\rho_0}P_y + F_y + \frac{1}{h^2}\frac{\partial}{\partial z}\left(\nu_v\frac{\partial v}{\partial z}\right) + M_y \tag{3}$$

where $f$ is the Coriolis parameter, $\rho_0$ is the water density, $P$ refers to the baroclinic pressure terms, $F$ is the imbalance of the horizontal Reynolds stresses with $\nu_v$ being the vertical eddy viscosity, and $M$ are the contributions due to short wave stresses. Under the assumption of shallow water, the vertical momentum equation is reduced to a hydrostatic pressure equation. Transport phenomena are modeled by solving the advection-diffusion equation for salt and heat transport and an equation of state to calculate the water density. The conservative form of the advection-diffusion equation (Lesser et al., 2004):

$$\frac{\partial(hc)}{\partial t} + \frac{\partial(huc)}{\partial x} + \frac{\partial(hvc)}{\partial y} + \frac{\partial(wc)}{\partial z} = h\left[\frac{\partial}{\partial x}\left(D_H\frac{\partial c}{\partial x}\right) + \frac{\partial}{\partial y}\left(D_h\frac{\partial c}{\partial y}\right)\right] + \frac{1}{h}\frac{\partial}{\partial z}\left(D_v\frac{\partial c}{\partial z}\right) + S \tag{4}$$

where $c$ is the concentration of the conservative constituents considered, $S$ is the source and/or sink term per unit area, and $D_h$, $D_v$ are the horizontal and vertical diffusivity coefficients, respectively. To solve this equation, Delft3D models use a cyclic method where viscosity and diffusivity must be prescribed as calibration parameters. The vertical eddy viscosity $\nu_v$ and the diffusivity coefficients $D_h$ and $D_v$ are obtained using (1) the contribution of a user-defined background value used for calibration, which can be spatially constant or variable; and (2) a 3D turbulence contribution obtained from a turbulence closure model. Density, salinity, and temperature are related using the UNESCO (1981) equation of state formulation. Heat exchange at the water surface is modeled considering the effects of solar (short wave) and atmospheric (long wave) radiation, as well as heat loss due to back radiation, evaporation, and convection. Among the available heat exchange models within Delft3D, the ocean heat flux model (Gill and Adrian, 1982; Lane, 1989) was used as it was found to be the most balanced option considering the dimensions of the bay and the temporal scale of the analysis (1 year). The model takes into account relative humidity, air temperature, cloud cover, shortwave solar radiation, light attenuation (Secchi depth), and the Dalton and Stanton numbers, the latter two being used as calibration parameters.

The short-wave propagation processes are solved in the model using the Wave module. This module is based on the SWAN model (Booij et al., 1999; Ris et al., 1999), a spectral wave model that solves the action balance equation taking into account wave-current interactions. It considers wave generation by wind and energy dissipation by whitecapping, wave breaking, and bottom friction. Both modules (Flow and Wave) are coupled online during the simulations: water levels and currents are considered for the wave propagation processes, while wave-induced forces are included in the momentum equations (eqs. 2 and 3).

### 3.2.2 Model set-up

For the hydrodynamic module, water level boundary conditions were defined as astronomical forcing using the 18 major tidal harmonics in the area (M2, S2, SA, Q1, O1, P1, K1, 2N2, MU2, N2, NU2, L2, T2, R2, K2, MN4, M4, and MS4) obtained from the Oregon State Tidal Prediction Model (Egbert y Erofeeca, 2002). For the transport boundary conditions, 3 h-interval data of salinity and temperature from buoy 2342 (Puertos del Estado, Spanish Ministry of Public Works; Fig.1c) were used. Despite their small influence, the average discharges of the two main rivers (Guadalete and San Pedro) were included as source terms, taking into account their temperature and salinity (data provided by the Andalusian Regional Government). The ocean heat flux model was fed with spatially uniform daily averaged data of solar radiation, air temperature and humidity, and cloudiness

(also obtained from the Regional Government of Andalusia). Finally, spatially uniform 1 h-interval wind data from buoy 2342 were used to estimate the wind contribution to the momentum equation. For the model calibration, the background roughness was set to a uniform value (Chézy $u = 80$, $v = 60$) on the whole grid without differences between zones, since Zarzuelo et al. (2018) observed that including the spatial variability of this coefficient did not improve the quality of the calibration. Spatially uniform values were used for the user-defined horizontal and vertical viscosity coefficients, and the K-epsilon model was used for the turbulence closure model, assuming a boundary layer type of flow. Still water and null heat fluxes were used for the initial conditions, with a spin-up interval of one month to warm up the model.

For the short wave propagation module, spectral offshore boundary conditions were used with 3 h-interval data from Buoy 2342. Wind data from this buoy with the same interval were used as spatially uniform forcing to consider the energy exchange between wind and short waves. The wave spectrum definition was limited to 24 frequencies between 0.05 and 1 Hz with 36 specified directions (15° steps). Physical processes such as whitecapping (Komen et al., 1984), depth-induced wave breaking (Battjes and Janssen, 1978), bottom friction (Hasselmann et al., 1973) and triads (Eldeberky and Battjes, 1996) were taken into account, but quadruplets were neglected.

The model domain covers the Atlantic Ocean from Rota to Sancti-Petri, including the Bay of Cádiz, the two freshwater rivers (San Pedro and Guadalete) and the tidal creeks (Carracas and Sancti-Petri) (Fig.1a-orange line). The grid defined for the hydrodynamic simulations has a spatial resolution ranging from 10 x 10 m in the area of the creeks to 5 x 5 m at the offshore boundary with a total of 90405 nodes. The vertical structure of the grid was implemented using a sigma-layer approach, for which the layer thickness is defined as a constant proportion of the total water depth. In this case, 10 layers with thicknesses of 2, 10, 10, 10, 10, 10, 10, 10, 10 and 2 % of the water depth were used. This number of layers was proven to be sufficient to capture vertical processes, as shown by Zarzuelo et al. (2018, 2021), since stratification is not important in bay dynamics. The grid for the wave module was based on the hydrodynamic grid, but with a lower resolution in creek areas where the role of waves is negligible (Zarzuelo et al., 2020), allowing a more efficient numerical scheme.

The offshore bathymetric data were provided by the Instituto Hidrográfico de la Marina (Spanish Ministry of Defense), while the detailed multibeam 2011 bathymetry of the bay was provided by the Bay of Cádiz Port Authority. The bathymetry of Sancti-Petri and Carracas Creeks was corrected with nautical charts. Finally, the topography was obtained from the 2010 digital terrain model of the Andalusian Regional Government with a resolution of $10 \times 10 \text{ m}^2$, combined in the intertidal zones with 2015 LIDAR data with $2 \times 2 \text{ m}^2$ resolution.

The results of the numerical model were extracted in a total of 553 observation nodes distributed along the bay (Fig.1b-empty circles). To capture the variability, the results were obtained by working only with history files (nodes) but defining a large number of observation points and a high temporal resolution of the output data. The variables included water level, currents, density, significant wave height, peak period and wave direction.

### 3.2.3 Calibration and testing

The model calibration and testing were performed using the data from the mounted fields 2012 and 2013 surveys (Field2012 and Field2013, Table 1), respectively. Waves, currents (residual and instantaneous), water levels, temperature, and salinity

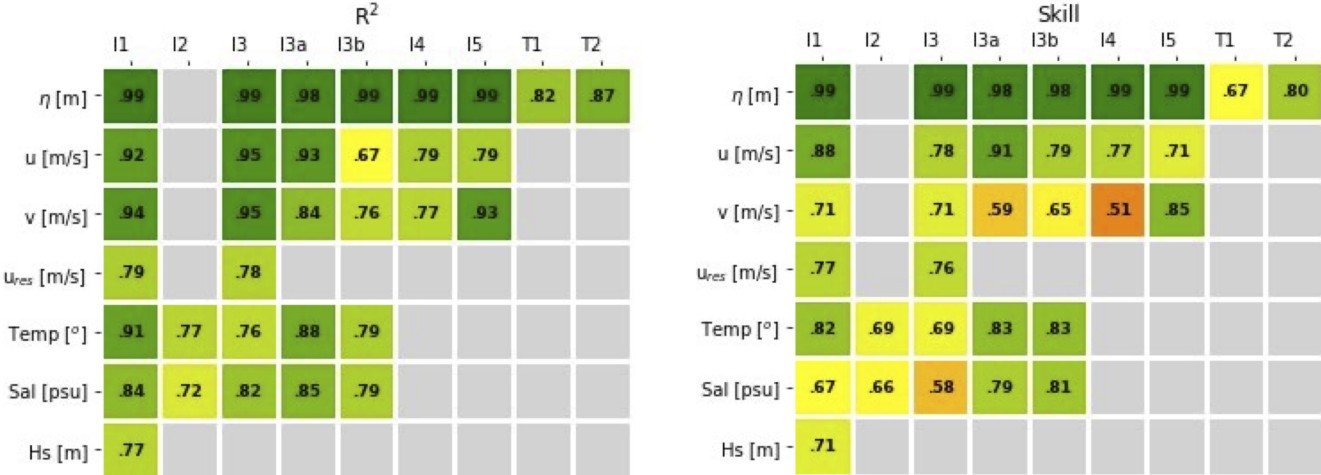

**Figure 2.** (a) Correlation coefficient ($R^2$) and (b) skill coefficient (formulation proposed by Willmott (1981)) of the water level, east and north instantaneous velocity, residual velocity, temperature, salinity and wave height for each station. The color indicates the degree of accuracy (green excellent agreement, yellow–orange good agreement and red poor agreement).

were analyzed by means of correlation and skills coefficients. In the case of temperature, salinity and instantaneous currents,
these variables were tested for both the 3D and depth averaged values. All measurements were checked visually and corrected
for outliers and suspect data points whenever possible. The 9 locations where field and numerical data were compared are
shown in Fig.2; full validation of the DELFT3D modeling approach is documented in Zarzuelo et al. (2015, 2020, 2021). Fig.2
depicts the correlation and skill coefficients for water levels, east and north velocities, residual currents projected on the axis of
the channel, temperature, salinity and significant wave height. Skill coefficients score is calculated following the formulation
proposed by Willmott (1981), to know in more detail the adjustment of the trend and the high and low peaks (Olabarrieta et al.,
2011; Zarzuelo et al., 2015).

    Fig.3 shows the agreement of the temperature and salinity, the two variables with the lowest correlations, between the
measured and modeled results. It can be seen that the mean values are properly captured by the model, while the outliers reduce
the agreement. The histograms show which are the most frequent values at the observation point for both the model and the
observed data. In the case of temperature, the most common value is 22°C for both the observed data and the model. However,
the distribution is more symmetric and accentuated for the observed data, whereas it follows a less skewed distribution for the
modelled data. In the case of salinity, the histogram and distribution follow a very similar trend in the observed and modeled
data, with the most common value being 36.5-36.6 psu.

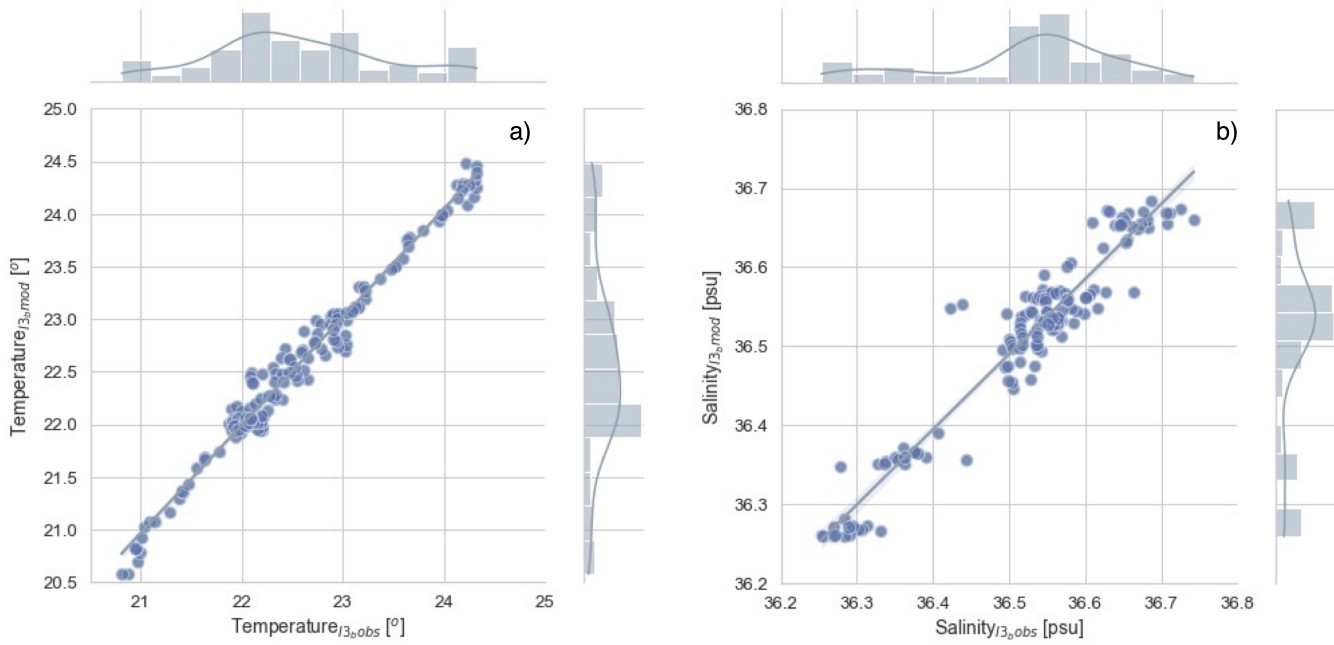

**Figure 3.** Test results of the temperature (a) and salinity (b) for location I3b (Field survey2013). The histograms and distributions of the observed data are shown at the top of the panels, and the histograms and distributions of the modeled data are shown at the right of the panels.

## 4 Data overview

The hybrid dataset obtained by applying the methodology described above has been used by the authors of this paper for different research applications, such as an analysis of the exchange flow between the main areas of the bay (Zarzuelo et al., 2022a), a characterization of seasonal dynamics and the influence of the sea-water heat exchange on bay dynamics (Zarzuelo et al., 2020), and an assessment of the impacts of recent human interventions on the estuarine dynamics (Zarzuelo et al., 2022a). In this section, an analysis of the spatial, fortnightly and seasonal variabilities is demonstrated as an example of the capabilities

and potential of the hybrid dataset. In addition, storm surges recorded during the winter of 2013 (higher winds) were selected to analyze the relationship of wind intensity-direction with water density changes.

### 4.1 Data record

Representative stations where the numerical results were obtained with a high time resolution (10 min) were used to analyze the bay dynamics from intratidal to seasonal scales (Fig.1). The length of the records for the different stations, together with

345 the variables measured, is shown in Fig.4. Some data gaps for the variables measured with the ADCPs (water levels, flow and wave data) occurred at the I1, I2 and I3 locations. During these gaps, only density measured with the CT instruments was recorded. The data gaps were caused by malfunctions or maintenance of the instruments.

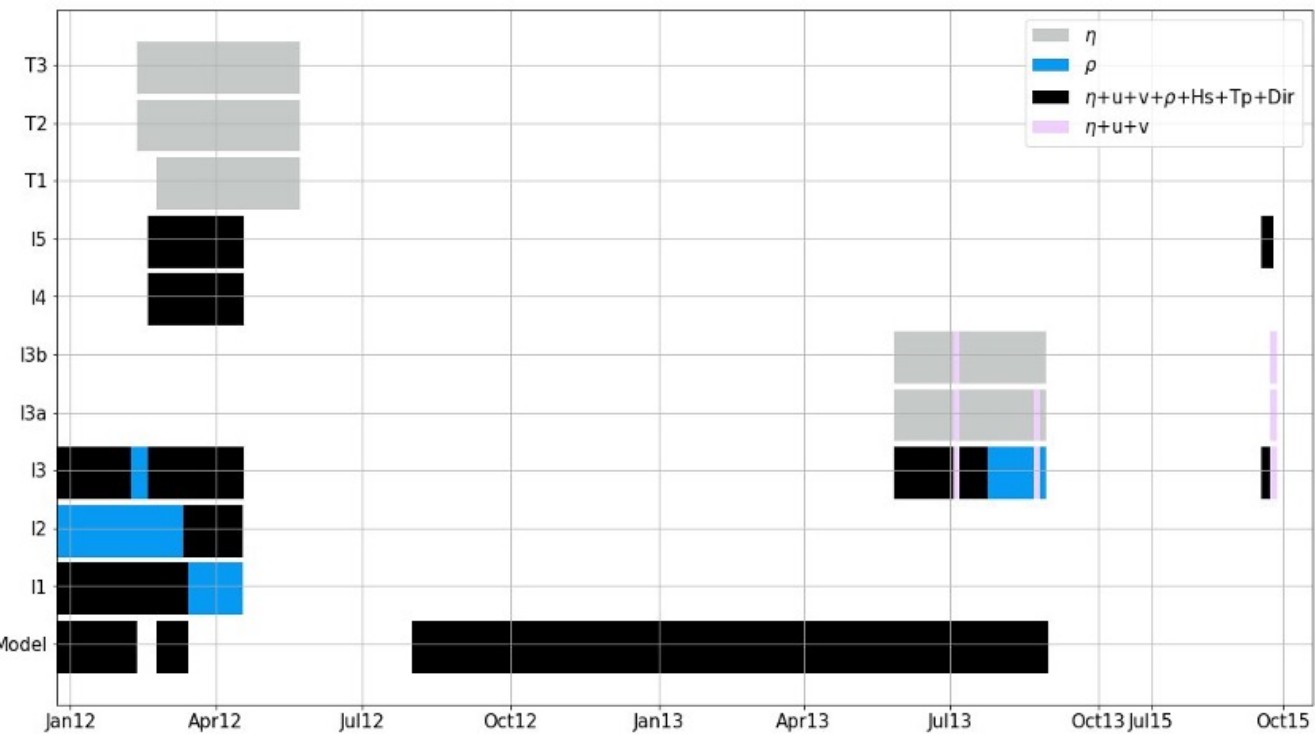

**Figure 4.** Extent of valid data recorded by the monitoring station and the numerical model during all field surveys. Vessel-towed observations are shown in purple (water level and velocity data from AQP). Bottom-mounted observations are shown in black (water level, velocity, density and wave data from ADCP and CT), blue (density data from CT) and gray (water level data from tide gauge and ADCP (velocity sensor failure)). Numerical model data are shown in black color (water level, velocity, density and wave data).

## 4.2 Spatial variability

Fig.5 depicts the results from the analysis of the spatial variability of the main tidal and wave characteristics along the bay.
The amplitude (Fig.5b) and phase (Fig.5c) of the main tidal constituent (M2) were calculated via a harmonic analysis of the depth-averaged flow velocities obtained at each station of the transect defined in Fig.5a using the numerical model for the 13-month simulation in 2012-2013 (Fig.4). Fig.5d shows the maximum significant wave height obtained with the same simulation at every station. Regarding the flow velocities, for the M2 amplitude, the highest values (1.5 m/s) are found where the Puntales channel connects with the inner basin, decreasing to almost zero (25-32 km) at the connection between the Carracas and Santi-
Petri Creeks. In addition, an increase of 0.5 m/s is observed just at the entrance of the Puntales channel (10 km) and the tidal channels (21 km). These variations, related to the convergence and divergence processes of tidal wave propagation across the bay, are resolved by the spatial and temporal resolutions of the numerical model results.

The tidal phase of the M2 constituent is related to the celerity of the tidal wave and how it changes during its propagation through the bay. The period of constituent M2 (12.42 hours) is the time required for the phase to complete a 360° cycle. Thus,

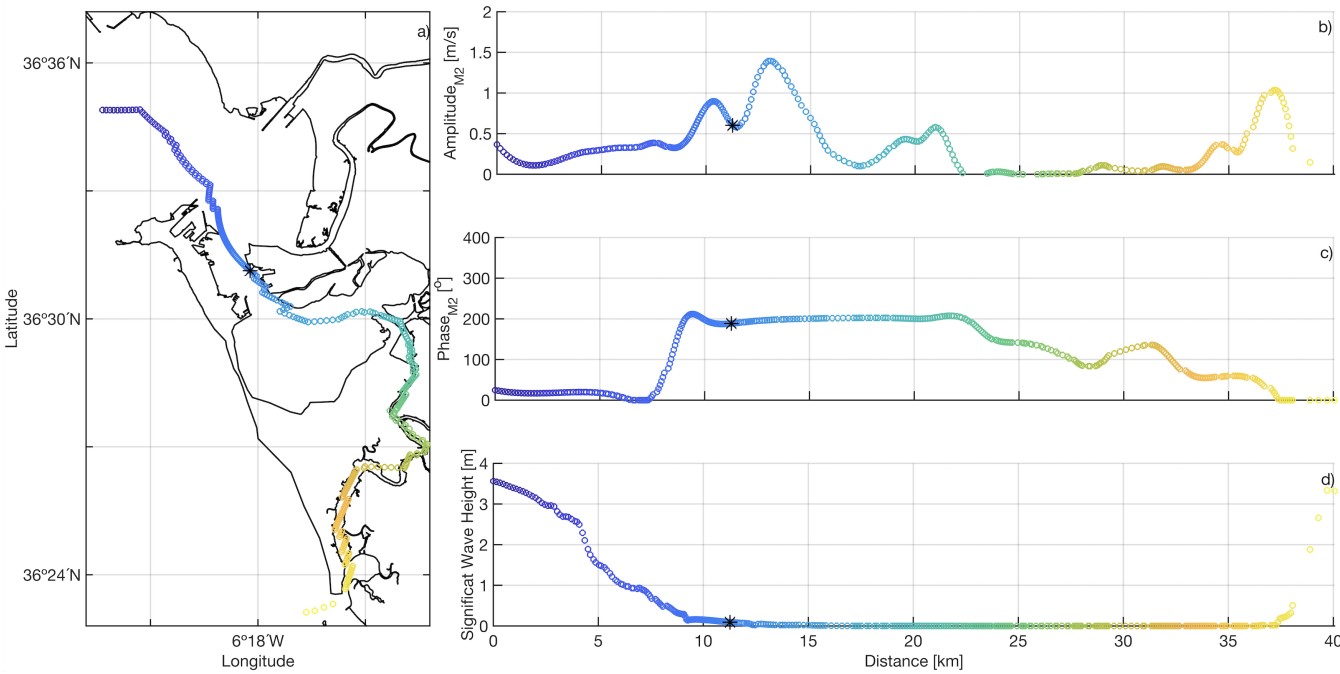

**Figure 5.** Panel a) represents the location of the points along the channel. The amplitude and phase of the tidal current constituents of the M2 along-channel are represented in Panels b) and c), respectively. Panel d) shows the variation in the significant height wave along the channel. The dot color indicates the distance. The asterisk corresponds to the location of the I3 instrument.

a phase of 200° corresponds to a wave tide propagation time of 6.9 hours. When the tidal wave enters the Puntales channel, the phase increases dramatically by 100° (3 hours). Then, the phase remains constant from the inner basin to the entrance of the creeks, from which it decreases almost linearly (from kilometer 22 (green-yellow color-Fig.5c)). It is also interesting to highlight the change that occurs around kilometer 30, since, as observed in previous studies (Zarzuelo et al., 2019), there is a point in the interior of the channels where the wave slows down. Finally, Fig.5d shows how wind waves dissipate when

propagating from the two connections of the bay with the open ocean to its interior, where their contribution is negligible in the Puntales channel, the inner basin and the Carracas and Sancti-Petri Creeks. From these results it can be concluded that the effect of the swell in the Bay of Cádiz is negligible, except in the outer part (Zarzuelo et al., 2020), and that the constriction serves as an amplifying element of the tide, which then dissipates into its interior (Zarzuelo et al., 2017).

## 4.3 Fortnightly variability

The hybrid dataset also provides insights into the fortnightly variability of the semidiurnal species along cross-sections of the Puntales channel. This type of analysis is useful to assess the tidal prism or water exchange between the two basins connected by this channel. Fig.6 depicts the amplitude of the semidiurnal species (D2) for the horizontal velocity of the flow obtained during the Neap2013 and Spring2013 surveys. The values are more than twice as high during spring tides, with maximum

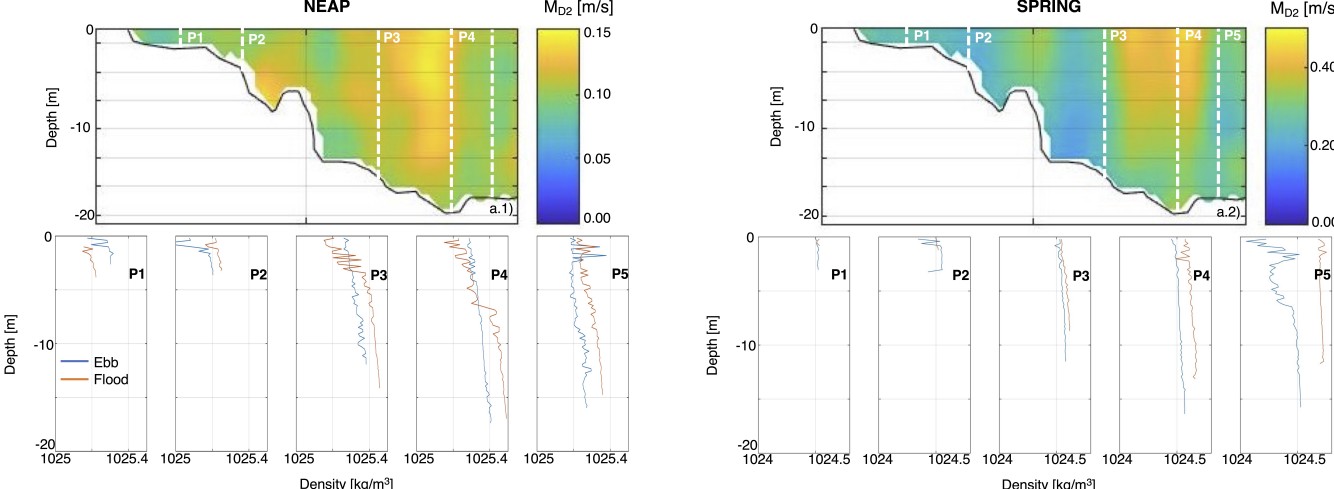

**Figure 6.** Amplitude of the tidal harmonic species D2 of the current during Neap2013 (a.1) and Spring2013 (a.2). The second rows show the density profile (P1-P5) during Neap2013 and Spring2013 at maximum flood (orange line) and ebb (blue line). The location of each profile is indicated by white dashed lines in a.1 and a.2.

values in the deepest zones (0.15 and 0.4 m/s for neap and spring tides, respectively). If we analyze the density profiles
measured during the same surveys, it is worth noting the decrease in density (0.4 kg/m$^3$) during the flood period, possibly coming from the seawater of the ocean (inverse estuary, as we can see in Fig.7b). This analysis is relevant to understanding the change in density as the tide propagates. The density and the tide determine the location of the pycnocline. When the tide changes from flood to ebb, the location of the pycnocline is closer to the ocean and as it changes from ebb to flood, it modifies the position of the pycnocline towards the inner basin. This is difficult to detect from field measurements, which only provide
cross-sectional information, but can be accurately determined from the model.

## 4.4 Seasonal variability

Of all the points obtained from the numerical model, 5 points were selected (Fig.7a) to facilitate representation. The density and residual water level time series data are displayed in Fig.7b and Fig.7c, respectively (October 2012– October 2013), together with observed records of the CT and ADCP located at I3. At this yearly scale, seasonal temperature variations are expected.
Pronounced and rapid density variations in summer are induced by the alternation of warm westward countercurrents that characterize ocean circulation. In addition, it is important to note that in the winter months, the inner basin (point 4-cyan line; Fig.7b) is denser than the outer basin (point 1-purple line; Fig.7b) (from 1027 kg/m$^3$ to 1029 kg/m$^3$). This tendency changes in summer, when the inner basin becomes less dense (from 1026 kg/m$^3$ to 1024 kg/m$^3$). Subtidal variability in the numerical data is well evidenced at a fortnightly time scale (Fig.7c). Larger density values are also clearly observed at spring tide for locations
1, 2, 3 and 4.

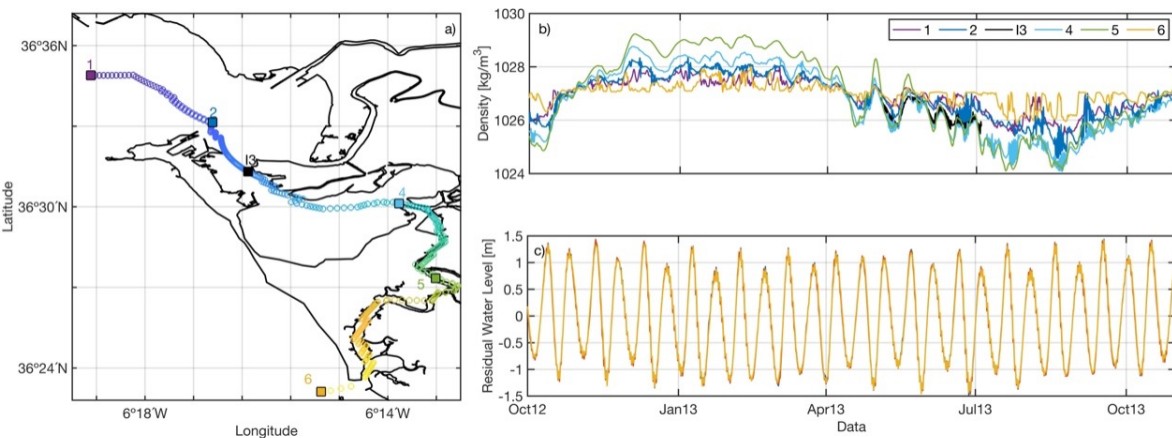

**Figure 7.** Panel a) represents the location of 5 points selected to show the seasonal variability of the density (b) and the residual water level (c). The data observed for I3 are plotted as a black line.

## 4.5 Extreme weather events

A wide range of weather conditions were captured during the measurements; in this section, we selected extreme wind events, as we observed in Zarzuelo et al. (2021) (winter period, Fig.8), to analyze how the density pattern changes according to the intensity and direction of the wind inside the bay. The same transect of Figs. 5 and 7 is selected for the first 20 kilometers
that coincide with the entrance to the creeks. Two well-marked density patterns are identified during this winter period. The first pattern can be observed when the density remains practically constant throughout the bay, coinciding with wind velocities below 5 m/s. However, the second pattern corresponds to an increase in density of 2 kg/m$^3$ between the outer and inner basins (orange rectangle, Fig.8) that is observed when mean wind velocities are approximately 10 m/s, and the incoming directions are from the north (1,2,4) and west (3).

As seen in Fig.7b, the estuary is an inverse one with denser seawater during winter; winds from the north and northwest introduce denser water into the inner basin. However, for the rest directions of the wind, the bay is completely well mixed. In addition, changes in the density pattern may be related to other variables, such as water level (Fig.8c) or increased precipitation/radiation (Fig.8d). During neap tides, the tides help to mix the bay and create homogeneous longitudinal profiles. Spring tides, however, cause differences in density throughout the bay. In this case, it can be observed how the increase in rainfall
causes an increase in the density of the water, which is more noticeable in the inner basin due to the resilience time.

## 5 Data availability

The data presented in this article are freely available at the ZENODO repository. See https://doi.org/10.5281/zenodo.7484187 (Zarzuelo et al., 2022b). The datasets are published in NetCDF format (.nc) with the observed and modeled results. The

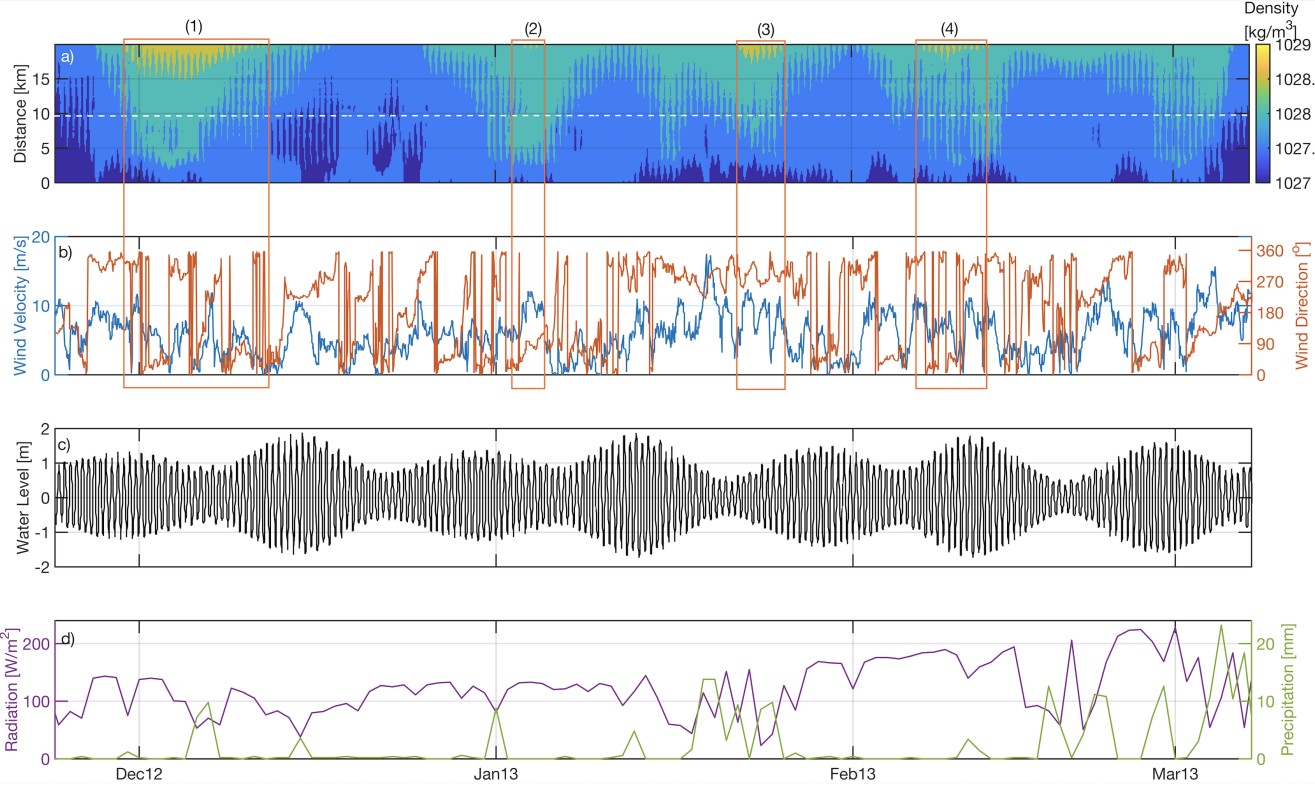

**Figure 8.** Panel (a) corresponds to the density variability along extreme wind events (winter period), and Panel (b) represents the wind velocity (blue line) and direction (orange line). The white dashed line corresponds to the location of instrument I3. The orange rectangle represents density changes throughout the bay. Panel (c) and Panel (d) represent the water level, and radiation (purple line) and precipitation (green line) recorded in the Bay of Cádiz, respectively.

observed data files (water level, current, density and wave climate) are referred to with explicit file names and include extensive
information (in header) about the site, instruments, setup and units.

The probe time series data are organized into five data files depending on the field survey:

  – Mounted Field2012: 22 December 2011 to 20 April 2012

  – Mounted Field2013: 15 May 2013 to 22 August 2013

  – Mounted Field2015: 16 September 2015 to 22 September 2015

– Neap2013 and Spring2013: Neap tide 7 July 2013 and Spring tide 22 August 2013

  – Neap2015: Neap tide 22 September 2015

The numerical data files (water level, water current, wave climate and density) are organized into several data files from September 2012 to October 2013 to the points described in Fig.1.

## 6 Conclusions

A unique and comprehensive dataset is presented, containing bathymetric, water level, current, density and wave climate data from the Bay of Cádiz (southern Spain). The data have a high spatial and temporal resolution and capture a wide range of climate conditions in the bay, including storm events. The high-frequency data are suitable for testing intrawave-scale models, where the spatial coverage allows comparison with larger-scale wave-resolving models.

This high-resolution dataset allows the analysis of intrawave processes in this complex environment, including the influence of tidal currents on wave transformation (de Wit et al., 2019). Different applications can be extracted due to the measured and modeled variables. As seen in Section 4, application can provide the following: (1) an analysis of the spatial variability of the main tidal harmonic and the variability of the significant wave height along the bay, (2) the seasonal variability of the density along the bay and (3) an analysis of how the density patterns vary according to wind events or according to the variability of the water level or other variables such as radiation or precipitation, among others.

The dataset can evaluate other variables or applications as a result of the obtained parameters. For example, these data provide the opportunity to analyze the influence of wind, waves and tidal flow on bed shear stresses, which are important for sediment transport. Moreover, the combination of ecological and physical data can be used to develop and verify (conceptual) models that describe interactions between biotic and abiotic processes. Ultimately, the dataset can be used to assess different scenarios such as future sea level rise and to support global warming mitigation and adaptation strategies.

*Author contributions.* CZ contributed to article compositions, article figures, article concept, maintain the station, analyse and organize the data sets, set up numerical modeling of DELFT3D and testing of DELFT3D. ALR contributed to set up numerical modeling of DELFT3D, model result processing and analysis, and lineage design. MB contributed to project initiation, article compositions, article figures, proofreading and article figures and concept. MOS contributed to directed the monitoring activities project initiation, supervising, proofreading and the article concept.

*Competing interests.* The authors declare no competing interests.

*Acknowledgements.* This work has been supported by the Spanish Ministry of of Economy and Competitiveness, PID2021-125895OA-I00 (RESILIENCE), and by Department of Economy, Knowledge, Business and Universities of the Andalusian Regional Government (Project A-TEP-88-UGR20). M. Bermúdez gratefully acknowledges funding from FEDER/Junta de Andalucía-Consejería de Transformación Económica, Industria, Conocimiento y Universidades: Project B-TEP-110-UGR20 and from EU's Horizon 2020 Programme under Marie Skłodowska-Curie Grant Agreement 754446 and UGR Research and Knowledge Transfer Fund—Athenea3i. Two anonymous reviewers are acknowledged for their comments and suggestions which improved significantly the manuscript.

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

| Field Survey | Location | Instrument | Height above bed | Settings | Wave measurements |
|---|---|---|---|---|---|
| Field2012 | I1 | ADCP (2MHz) + OBS | 0.5 m | 2 min bursts every 15 min, bin size 0.5 m | 17 min bursts every 120 min, bin size 1 m, 1 Hz |
| | | CT | 0.3 m | 1 sample every 15 min | |
| | I2 | ADCP (1MHz) | 0.7 m | 2 min bursts every 15 min, bin size 0.75 m | 17 min bursts every 60 min, bin size 1 m, 1 Hz |
| | | CT | 0.3 m | 1 sample every 15 min | |
| | I3 | ADCP (1MHz) | 0.7 m | 2 min bursts every 15 min, bin size 0.5 m | 17 min bursts every 60 min, bin size 2 m, 1 Hz |
| | | CT | 0.3 m | 1 sample every 15 min | |
| | I4 | ADCP (2MHz) | 0.5 m | 2 min bursts every 15 min, bin size 0.5 m | 17 min bursts every 60 min, bin size 1 m, 1 Hz |
| | I5 | ADCP (2MHz) | 0.5 m | 2 min bursts every 15 min, bin size 0.5 m | 17 min bursts every 60 min, bin size 2 m, 1 Hz |
| | T1 | Tide Gauge | 0.1 m | 10 sec bursts every 15 min, 1Hz | |
| | T2 | Tide Gauge | 0.1 m | 10 sec bursts every 15 min, 1Hz | |
| | T3 | Tide Gauge | 0.1 m | 10 sec bursts every 15 min, 1Hz | |
| Field2013 | I3 | ADCP (0.6MHz) | 1.3 m | 1 min bursts every 30 min, bin size 1 m | 17 min bursts every 240 min, bin size 4 m, 1Hz |
| | | CT | 0.6 m | 1 sample every 15 min | |
| | I3a | ADCP (2MHz) | 1.0 m | 1 min bursts every 15 min, bin size 1 m | 17 min bursts every 120 min, bin size 1 m, 1 Hz |
| | | CT | 0.6 m | 1 sample every 15 min | |
| | I3b | ADCP (2MHz) | 1.0 m | 1 min bursts every 15 min, bin size 1 m | 17 min bursts every 120 min, bin size 1 m, 1 Hz |
| | | CT | 0.6 m | 1 sample every 15 min | |
| Field2015 | I3 | ADCP (0.6 MHz) | 1.3 m | 0.5 min bursts every 5 min, bin size 1 m | 17 min bursts every 60 min, bin size 4 m, 1 Hz |
| | | CT | 0.6 m | 1 sample every 5 min | |
| | I5 | ADCP (1MHz) | 1.2 m | 0.5 min bursts every 5 min, bin size 1 m | 17 min bursts every 60 min, bin size 2 m, 1 Hz |
| | | CT | 0.6 m | 1 sample every 5 min | |

**Table 1.** The mooring deployments for bottom-mounted surveys during the three field surveys: 2012, 2013 and 2015.