# Peer review of "Measurements and modeling of water levels, currents, density and wave climate on a semi-enclosed tidal bay: Cádiz (SW Spain)"

_Earth System Science Data, 2022_

## Author Response (AR1)

ESSD-2022-467

**AUTHORS'S RESPONSE TO REVIEWERS**

**Measurements and modeling of water levels, currents, density and wave climate on a semi-enclosed tidal bay: Cádiz (SW Spain)", by Carmen Zarzuelo et al.**

**Referee #1**

*In this paper the authors present an oceanographic dataset comprised by observations and numerical model outputs, covering periods of different lengths between 2012 and 2015. Observations are divide into fixed stations measurements and vessel surveys. The stations were equipped with ADCP, CT probes and punctually with a turbidity sensor. The vessel observations consist of CTD profiles, including fluorometer and turbidometer measures, water samples and velocity measures from the vessel-mounted ADP. The simulation is performed by an implementation of the Delft3D model with very high resolution (between 10 and 5 meters), which includes tidal and wave forcing in the boundaries, as well as local river discharge. The model was validated using in situ observations. In addition to the data description, examples of the capacity of the combined dataset to describe processes at different time scales are shown and analyzed.*

*In my opinion, the dataset presented has a great potential for many different kind of studies in the area, from climatic and ecological analysis to engineering projects, as mentioned by the authors. However, the description of the dataset and the presentation of the application examples given in the paper should be improved before considering its publication in ESSD. Therefore, my recommendation is a major revision at least of the points I list below.*

The authors appreciate the careful review and the suggestions for improvement of the manuscript. Each comment is addressed in detail below.

**General points**

*1) There are two major aspects that need a thorough review. First, the description of the dataset, particularly the observations. In order to understand and use the dataset, potential user should have a detailed description of all the instruments used in the fixed locations and in the vessel. The description given in section 3.1.1 and 3.1.2 is very shallow, and the information summarized in table 1 is insufficient to properly understand the data. You should include, at least, the model and main characteristics of the instruments used, their calibration, accuracy, uncertainty, etc. From table 1, one assumes that different instruments have been used for the different sites and periods, but no information is given about them.*

The observed data were not described in sufficient detail in the original version of the manuscript. Following the reviewer's suggestions, sections 3.1.1 and 3.1.2 have been thoroughly revised (see manuscript with marked changes) and the requested information has been included. Specifically, we have included: (1) the model, resolution, and accuracy of each instrument used; (2) a more detailed description of the time periods and locations of each field survey; and (3) the maximum range of each variable recorded by the instruments. To improve the readability of this information,

we have modified Table 1 by subdividing it into soundings and their locations, and by indicating the acoustic frequency of the ADCPs and the configuration of each instrument.

*2) In the post processing section, you don't give any quantitative description of the data quality. How many measures were discarded? Is the dataset homogeneous?... Also, you mention that the upward looking ADCP measures the whole water column, but how do you deal with the side lobe interference of the instruments? Usually, the samples in the last cell of the ADCP record are strongly affected by spurious signals reflected from the surface. This effect depends on the working frequency of the instrument, which in your case varies among location and time periods. How have you solved this problem?*

The authors acknowledge the reviewer's comment. With regard to post-processing, further specifications have been included on how this was done and the amount of data and information discarded. In addition, the calibration of variables has been specified depending on the instrumentation used. The blanking performed on the last cells is to remove information above the air-water interface (~20%, depending on whether it is high or low tide). And with regard to the data removed by the software, taking into account the SNR side-lobe rejection, std, tilt effects and echo peaks are around 12%, leaving 88% of the ADCP validated data. The percentage of values removed by the software is an estimate as there is little control over this. These changes have been incorporated in section 3.1.3 of the revised manuscript (see manuscript with marked changes).

*3) My second general concern is about the presentation, particularly the figures. Figures should be easily readable, and as self-contained as possible. Most of the figures captions don't mention all the elements shown in the figures. Color scales and axes are sometimes different from one panel to the another, so comparisons are difficult and some lines and other elements are very difficult to see. A thorough review of all the figures and figure captions is needed. Also, a review of the text is also necessary, there are some words and structures that are incorrect or difficult to understand. When analyzing the examples, even though is not the objective of the paper to analyze them in depth, further explanation of some of the hypothesis made to explain the processes described and some references are missing.*

We appreciate the reviewer's comments on this point. Most of the concerns raised have been addressed by responding to the specific points. In addition, the figures and their legends have been thoroughly revised to make them more readable and self-explanatory. In addition, some of the results that were originally not explained in sufficient detail have been described in more detail so that the reader can easily follow the manuscript. In particular, Figures 1, 3, 5, 6, and 8 have been modified, as have the captions for Figures 1, 2, 3, and 4. The interpretation of Figure 3 has also been changed to make it easier to understand (see lines 361-367 of the marked version of the manuscript).

*I list bellow some specific points, but a general review of these aspects is necessary.*

**Specific points**

*L7: what do you mean by water levels?*
We refer to the elevation of the water surface above the mean sea level
*L82: sheltered → were/are carried out*
Done.
*L85: Give a reference for the values you mention.*
Done.

*Figure 1: The figure and caption should include clearly the model domain, to which period/campaign each site corresponds, what are the model data points. All colors, letters and numbers used should be explained in the caption with no abbreviations.*

All information has been included and explained in more detail in the figure.

*L133: how do you deal with the side lobe interference when sampling the nearest cell to the surface?*

This comment has been addressed in the response to the main comments.

*L134: Do you use different instruments? Which ones?*

We used three instruments in the mounted field survey: ADCPs, CTs and tide gauges. This information is given in the text and in Table 1 (see response to main comments for more details).

*L135: here 1Hz is the sampling frequency not the working frequency, am I right? Mention it please.*

We acknowledge the reviewer's comment, it is an error, the sampling frequency is different in each instrument and in each survey. This information has been clarified in Table 1 and in the text (see response to main comments for more details).

*L226-228: The model grid, or at least the limits of the domain, should be included in figure 1.*

Done.

*L239-242: This information should be clearer in figure 1.*

Done.

*L250: Which skill score do you use to evaluate the accuracy of the model? You should explain more about this. Potential users need to know on detail how do you evaluate the accuracy of the dataset. Same applies to the observations.*

The formulation proposed by Wilmott (1981) has been used. This coefficient is used to know in more detail the adjustment of the trend and of the maxima and minima. This information has been introduced in the text.

*Figure 2. Indicate the skill score used. Acceptance → agreement/accuracy/correlation.*

Done.

*Figure 3. Here a complete description of the figure is missing. What do you mean by test results? What are the histograms and distributions shown on the axes? This should also be more elaborated in the text (L252-254).*

Following the reviewer's comment, a description of the figure has been added to the revised version of the manuscript.

*L271-272: fig.2 → fig. 5*

Done.

*L278: revealed → resolved?*

Done.

*L282: I don't see the decrease you mention in the figure.*

The decrease refers to the reduction of the phase shown from kilometre 22 to the South (green-yellow colour). It has been included in the revised text.

*L282-285: any hypothesis on why you observe this?*

According to our previous work and studies in the area, we can conclude that the effect of the swell is negligible in the Bay of Cadiz, except in the outer part as seen in Zarzuelo (2020). Also, the constriction acts as an amplifying element of the tide before the decrease towards the inner part of the bay, as observed in Zarzuelo (2017, 2021).

*fig2d → fig. 5d*

Done.

*Figure 4. Here again, a more detailed description of the figure is missing. All the variables in the legend should be explained. Gray lines indicate the tide gauge and ADCP failures, meaning that for the rest of the time they were working?*

It has been included and explained with more detail in the revised version of the manuscript.

*L294-295: This sentence is not clear. Please explain with more detail what you mean.*

Done.

*Figure 5. Significant height wave → significant wave height.*

Done.

*Figure 6. Please explain all the elements in the figure. What are the vertical white lines (I see they correspond with the profiles below but you should mention it). Please use the same limits for the color bars and the density axes in all the panels. It would be much easier to compare them this way.*

The figure has been clarified by changing the caption. The same limits have been used for neap and spring tides, but their values are quite different and cannot be easily distinguished.

*L303-304: According to the figure is the opposite. The inner basin is denser in summer and lighter in winter.*

The inner basin corresponds to point 4 and the outer basin to point 1. In the winter months, point 1 has a density of 1027 and point 4 has a density of 1029, so the inner basin is denser. However, in summer, point 1 has a density of 1026 and point 4 has a density of 1024, so the inner basin is less dense. The text has been changed for better understanding.

*L307: for 1, 2, 3 and 4 → for locations 1, 2, 3 and 4.*

Done.

*L311-315: Former and latter are not correctly used here. Please rephrase.*

Done.

*Figure 8. The font in this figure is too small, very difficult to read. The orange rectangles are also very difficult to distinguish.*

This figure has been modified and improved according to the reviewer's comments.

*L316-319: Please elaborate more and rephrase. It is not clear what you mean. How do you relate the precipitation and radiation variability with the density changes? If you have a hypothesis you should explain it clearly.*

During neap tides, the tides help to mix the bay, resulting in homogeneous longitudinal profiles. During spring tides, however, there are differences in the density of the bay. In this case, one can observe how the increase in precipitation causes an increase in water density. This is more pronounced in the interior of the bay due to the resilience time. This information has been included in the revised version of the manuscript.

*L339-340: Please rephrase this sentence.*

Done.

**Referee #2**

*The paper introduces a database consisting of real data collected from six field surveys, along with the results obtained from a 3D numerical model of the Bay of Cádiz. This bay is an estuary with a complex geometry, comprising different interconnected basins through narrow channels. The study provides a characterization of the hydrodynamics, salinity, and temperature within the bay. The results offer insights into the spatial and seasonal variability of the estuarine dynamics and analyze the impacts of severe weather events.*
*The numerical simulations are based on a two-dimensional depth-averaged Shallow Water Equation, an advection-diffusion equation for heat and salt, and an equation for water density. These equations form the model upon which the simulations are built.*

The authors are grateful for the reviewer's overall assessment of the paper. They are also grateful for the specific comments below, which will undoubtedly improve the quality of the manuscript.

**Regarding some comments and questions:**

*It would be beneficial to explicitly include the model used for the numerical simulations. While it is briefly described in Section 3.2.1 on page 8, it would be helpful to provide specific formulae or equations to further clarify the model's implementation.*
As suggested by the reviewer, the mathematical description of the main governing equations of the model has been included in a new section (*Model Description*, Section 3.2.1 in the revised manuscript).

*Could you provide more information about the "K-epsilon model" with constant viscosity?*

The relationship between the turbulent closure model and the vertical eddy viscosity and the horizontal and vertical diffusivity coefficients has been described in detail in the new section mentioned above. The specific application of the k-epsilon model is described in the Model Setup section (3.2.2) of the revised manuscript.

*Overall, the work appears to be interesting, well-written, and potentially valuable for future investigations.*